# Data augmentation using image translation for underwater sonar image segmentation

**Eon-ho Lee**[1], **Byungjae Park**[2], **Myung-Hwan Jeon**[3], **Hyesu Jang**[4], **Ayoung Kim**[4], **Sejin Lee**[1]*

**1** Division of Mechanical and Automotive Engineering, Kongju National University, Cheonan, South Korea, **2** School of Mechanical Engineering, Korea University of Technology and Education, Cheonan, South Korea, **3** Robotics Program, KAIST, Daejeon, South Korea, **4** Department of Mechanical Engineering, Seoul National University, Seoul, South Korea

☯ These authors contributed equally to this work.

* sejiny3@kongju.ac.kr

**Data Availability Statement:** All relevant data are within the paper and its Supporting information files.

**Funding:** This research was supported by the National Research Foundation of Korea (NRF) grant

## Abstract

In underwater environment, the study of object recognition is an important basis for implementing an underwater unmanned vessel. For this purpose, abundant experimental data to train deep learning model is required. However, it is very difficult to obtain these data because the underwater experiment itself is very limited in terms of preparation time and resources. In this study, the image transformation model, Pix2Pix is utilized to generate data similar to experimental one obtained by our ROV named SPARUS between the pool and reservoir. These generated data are applied to train the other deep learning model, FCN for a pixel segmentation of images. The original sonar image and its mask image have to be prepared for all training data to train the image segmentation model and it takes a lot of effort to do it what if all training data are supposed to be real sonar images. Fortunately, this burden can be released here, for the pairs of mask image and synthesized sonar image are already consisted in the image transformation step. The validity of the proposed procedures is verified from the performance of the image segmentation result. In this study, when only real sonar images are used for training, the mean accuracy is 0.7525 and the mean IoU is 0.7275. When the both synthetic and real data is used for training, the mean accuracy is 0.81 and the mean IoU is 0.7225. Comparing the results, the performance of mean accuracy increase to 6%, performance of the mean IoU is similar value.

## Introduction

Recognition of objects underwater is essential for rescue or evidence search operations [1, 2]. However, cameras that are mainly used on land are difficult to use underwater for object recognition because the visibility is poor due to insufficient lighting and floats in water [3]. Unlike cameras, underwater sonar can be used in water because its signals can reach a long distance without being affected by the lighting or suspended solids [4, 5]. However, images obtained from imaging sonar are difficult to use for object recognition because their resolution is not high and they contain noises. Several object recognition methods have been proposed to

funded by the Korean government (MSIT) (No. 2019R1F1A1053708, 2021R1F1A1057949) and This research was supported by Development of standard manufacturing technology for marine leisure vessels and safety support robots for underwater leisure activities of Korea institute of Marine Science & Technology Promotion (KIMST) funded by the Ministry of Oceans and Fisheries (KIMST-20220567).

**Competing interests:** The authors have declared that no competing interests exist.

address this issue [6]. For example, a spectral analysis method has been proposed for seafloor sediment classification [7]. Another study proposed using a measure called lacunarity to classify the characteristics of the seafloor [8].

This paper expands on the results of the previous work and proposes a method that uses a neural network (NN) model for image translation [9] to synthesize realistic underwater sonar image (USI)s and then uses them in data augmentation. If image translation is used, it is possible to transform the style of the synthetic USI similar to that of the real USI. In the previous work, the variability is limited. In contrast, in this paper, the variability of the results of image translation is ensured, which is advantageous for data augmentation.

Furthermore, in the previous work, only the synthesis effect for the background noise can be expected in a limited manner. In contrast, in this paper, the gradation effect of the background noise and the object shadowing effect are produced like real ones by image translation according to the fan shape and location of the background noise, which are basic styles of multibeam imaging sonar images.

For the validation of the effectiveness of the proposed image translation-based data augmentation, we evaluated the quantitative performance of the semantic segmentation NN trained using the proposed method. Semantic segmentation can find not only the location of the object but also the shape in a given image by performing pixel-level classification. As semantic segmentation performs a more complex task than object classification or detection, the number of parameters of the NN for semantic segmentation is greater than that for object classification or detection. As more data are required to train the NN for semantic segmentation than other NNs, it is a good task for validating the effectiveness of the proposed image translation-based data augmentation.

The remainder of this paper is structured as follows. Section 2 briefly introduces the proposed image translation-based data augmentation and the pipeline that performs semantic segmentation for the verification of its performance. Sections 3 and 4 describe the image translation-based data augmentation and image segmentation, respectively. Section 5 introduces information related to the underwater sonar dataset and the training of the image translation NN and semantic segmentation NN. Section 6 presents the experimental results and qualitative performance evaluations, and Section 7 summarizes the conclusions of this paper and the future work.

## Related work

Recently, deep-learning-based object recognition methods have been suggested. For instance, a method that uses a convolutional neural network(CNN) to extract features and then uses a support vector machine to perform object classification has been proposed [10]. Furthermore, some researchers have proposed methods that apply an end-to-end approach while using a CNN to extract features for object detection or classification [11–13]. There is a critical limitation when deep-learning-based methods are used underwater for object recognition. NN, the most important part of deep learning, consists of numerous parameters, which are trained based on data. If data are not sufficient, the parameters of the NN are not properly trained; consequently, if overfitting occurs, a robust operation cannot be expected.

Unfortunately, it is a challenging task to collect data abundantly to train the NN due to the characteristics of the underwater environment. First, although the underwater environment is very large and has various characteristics, the region where data can be collected is limited. Second, considerable amounts of time and resources are required for data collection in the underwater environment. There are two solutions for training an NN with a small dataset: using transfer learning [14, 15] and data augmentation. Transfer learning method reuses a

pretrained model trained with a large dataset as a backbone of an NN for a specific task. The lower layers of the NN copies parameters of a pretrained model, and then the NN trained with a small dataset. Data augmentation refers to the transform of data in a dataset, such as a crop or resize [16]. If transfer learning and data augmentation using a transform are performed, the NN can be trained better; however, when the size of the dataset is small, these solutions may be only a supplementation within a limited range.

Some researchers have proposed a method that performs data augmentation by synthesizing data instead of data augmentation using a transform [16, 17]. This method creates synthetic data by transforming styles, such as the background patterns, as if the data have been obtained underwater. Then, the synthetic data are used with the real underwater data together when training the NN. In our previous work [18], we used a supervised style transfer to transform styles, such as the background patterns of a synthetic USI generated via simulation, into styles similar to those of the real USI; subsequently, we used them to augment the training data of the NN for object detection.

## Overview

As shown in Fig 1, the pipeline that performs underwater sonar semantic segmentation using the proposed data augmentation method consists of two stages overall: (1) the training of the image translation NN and the generation of a synthetic USI using the trained image translation NN; (2) the training of the semantic segmentation NN using both the real USI and the synthetic USI generated in the previous stage.

## Data augmentation using image translation

### Image translation model

The proposed method uses Pix2Pix, which is an image translation NN using cGAN [9, 19]. Pix2Pix consists of two sub-NNs: generator ($G$) and discriminator ($D$). $G$ generates a fake image $y$ for the input source image $x$ and the random noise $z$. $D$ distinguishes whether the input image is real or fake.

$$y = G(x, z). \tag{1}$$

During the training of the image translation NN, $G$ is trained to generate synthetic USI that are difficult to distinguish from real images. Simultaneously, $D$ is trained adversarially to distinguish real and synthetic USIs properly. In the proposed method, a synthetic USI image with

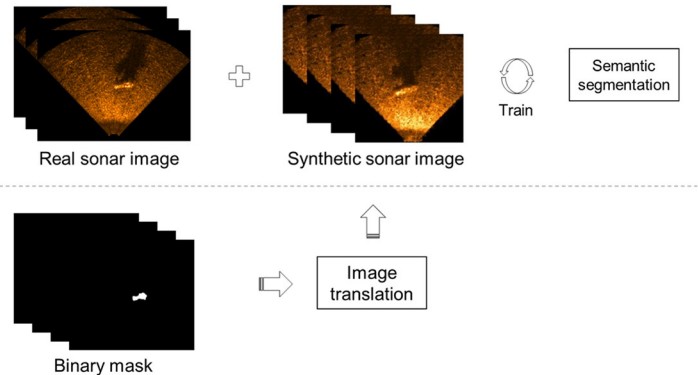

**Fig 1. Pipeline of the proposed method.**

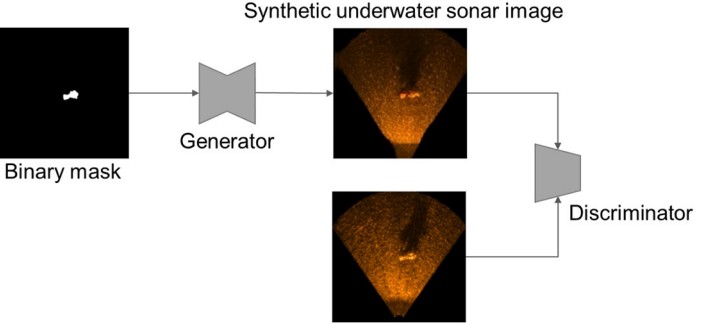

**Fig 2. Image translation NN used in the proposed method.** The generator generates a synthetic USI from the binary mask; the discriminator distinguishes the real USI and the synthetic USI generated by the generator. The generator and discriminator are trained together.

a binary mask is used as an input of $G$; the real USI and the synthetic USI generated by $G$ are used as the inputs of $D$ as shown in Fig 2.

$G$ has a structure similar to that of the U-Net model [20], in which multiple skip-connections exist in the encoder-decoder. Multiple skip-connections deliver contexts of multiple levels from the encoder to the decoder directly, contributing to improving the quality of the synthetic USI generated from $G$. $D$ uses a PatchGAN model [9]. It is used to prevent the phenomenon whereby the trend of updating the parameters of $G$ during the training of the image translation NN focuses more on deceiving $D$ than making the synthetic USI similar to the real USI. If this phenomenon occurs, $G$ generates a blurry synthetic USI. When the PatchGAN model compares the real USI and the synthetic USI, it does not compare the entire image but compares in patch units of certain regions. When the PatchGAN model is used, $G$ generates a sharper synthetic USI.

When the $G$ and $D$ of the image translation NN are trained simultaneously, the following two losses are used together: (1) adversarial loss; (2) L1 loss. The adversarial loss is used to train $G$ and $D$ simultaneously, which are two sub-NNs that have adversarial goals in cGAN [9]:

$$L_{adv}(G, D) = E_{x,y}[logD(x, y)] + E_{x,y}[log(1 - D(x, G(x, z)))]. \tag{2}$$

The L1 loss is used together to not only deceive D when updating the parameters of G in the training process but also generate synthetic USIs similar to the real USIs [9]:

$$L_{L1}(G) = E_{x,y,z}[\|y - G(x, y)\|_1]. \tag{3}$$

The final loss function can be defined as follows [9]:

$$G^* = arg \max_G \min_D L_{adv}(G, D) + \lambda \cdot L_{L1}(G), \tag{4}$$

In the above equation, $\lambda$ is a hyperparameter for controlling the influence of the L1 loss.

$G$ consists of 8 encoder blocks and 8 decoder blocks each. In each block of the encoder, a convolution layer is used, the method of normalization is a batch norm, and activation function is Leaky ReLU. In each block of the decoder, transposed convolution layer and dropout is used, a method of normalization is the batch norm, and activation function is ReLU. $D$ consists of 3 blocks. Each block uses a convolution layer, batch norm as a method of normalization, and Leaky ReLU as an activation function.

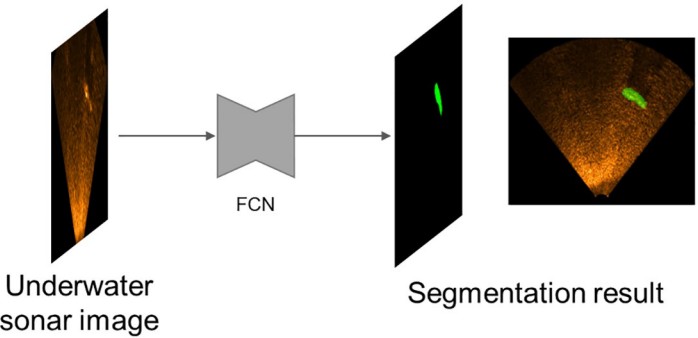

**Fig 3. USI segmentation using a fully convolutional neural network.**

## Underwater sonar image segmentation

### Image segmentation model

Semantic segmentation performs the task of predicting each pixel's class in a given image. Therefore, if semantic segmentation is applied to the USI, we can determine which pixels in the USI are occupied by the underground object that we want to find as shown in Fig 3. A fully convolutional networks (FCN) [21] is applied for the USI segmentation. A FCN is built by modifying VGG16 [22], a well-known NN used in image classification. The fully connected layers in VGG16 [22] are removed, and then $1 \times 1$ convolution layers, upsampling layers, and skip-connections [20] are added to facilitate dense prediction that outputs the pixel-level classification result with the same size as the input image.

## Underwater sonar image dataset

### Synthetic underwater sonar image generation

To train the image translation NN to generate synthetic USIs, we need a training dataset in which the source images and the images targeted to be generated are paired. To create the dataset, the real USIs are collected first, and then, the annotation tool is used to create the binary masks for the objects that will be segmented as shown in Fig 4.

After training the image translation NN, $G$ in the image translation NN is used to generate a USI as shown in Fig 5. The $G$ can synthesize USIs of the object with various poses and lighting conditions, which do not exist in the training dataset.

The dataset and the hyperparameters used to train the image translation NN will be described in detail in "Experimental Results".

### Training using synthetic underwater sonar image

The NN for semantic segmentation has a larger number of parameters than object classification and detection because it performs dense prediction. Therefore, a considerable amount of data is required to train the semantic segmentation NN. Moreover, the task of annotating the ground truth for semantic segmentation requires more effort than annotating the ground truth used for image classification or object recognition on an image. To make a ground truth, pixel-level labeling is required for semantic segmentation.

The real USI and the aforementioned USI synthesized using the image translation NN are used together to train the NN for USI segmentation. The following are the advantages of using a synthetic USI: (1) the effort required for additional experiments to obtain real USIs can be

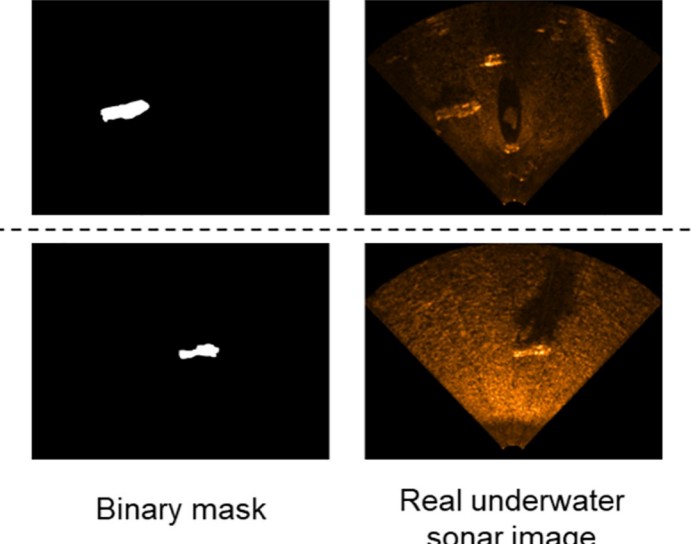

Binary mask          Real underwater
                     sonar image

**Fig 4. Example of a training dataset.** That has pairs to be used in the image translation NN training for USI generation.

reduced; (2) in the case of a synthetic USI, a task of annotating the ground truth is not required.

## Environmental conditions of underwater sonar image dataset

We constructed datasets in two actual underwater environments using TELEDYNE BlueView M900–90 sonar to train the image translation NN and the semantic segmentation NN. The sonar had a frequency of 900 kHz, a beam width of 90˚ in the horizontal direction and 20˚ in the vertical direction, and a detection range of 100 m.

The first underwater environment was a reservoir Fig 6A. In this environment, we sunk a mannequin to the bottom of the reservoir, and then attached the sonar to the boat to obtain

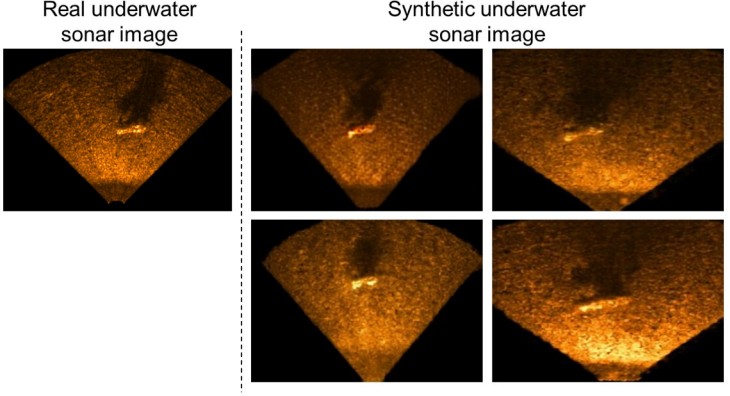

Real underwater          Synthetic underwater
sonar image              sonar image

**Fig 5. Real USI and USIs generated by the generator after training the image translation NN.** The generator of the image translation can create USIs of the object with various poses and lighting conditions, which do not exist in the training dataset.

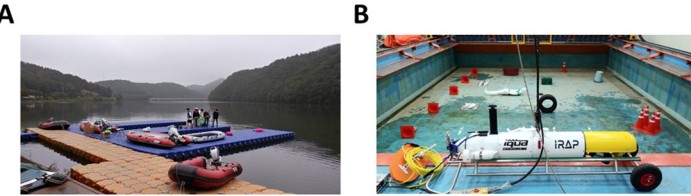

**Fig 6. Underwater environments and platforms.** (A) Reservoir and boat, (B) Pool and underwater vehicle.

the data. The second environment was a water pool Fig 6B. The pool had a width of 10 m, a length of 12 m, and a depth of 1.5 m. After sinking a mannequin with other artificial objects, such as a box and tire, to the bottom of the pool, we attached the sonar to SPARUS [23], an unmanned underwater vehicle, to obtain the data. When acquiring data in each environment, we adjusted the sensitivity of the sonar to create two conditions in each environment (reservoir high sensitivity—RHS, reservoir low sensitivity—RLS, pool high sensitivity—PHS, and pool low sensitivity—PLS) as shown in Fig 7.

NVIDIA T4 GPU and Keras package were used to train the image translation NN and semantic segmentation NN. We trained four image translation NNs to generate the synthetic USI for the datasets of the four conditions. In the training dataset for each environment, a binary mask and a USI are paired up, as shown in Fig 4, and there are 200 paired images with a size of 512 × 256. So, real USI used is a total of 800 pages. The synthetic USI generated is 200 pages for each type, for a total of 800 pages with a size of 640 × 480. The image translation NN was trained using the training dataset with Adam optimizer for 150 epochs, and the hyperparameters of the sub-NNs, *G* and *D*, were as shown in Table 1.

The semantic segmentation NN was trained to classify the pixels occupied by the object from the background in the USI based on the real datasets of the four conditions and the synthetic datasets of four conditions generated through the image translation NN. The training dataset for each condition consisted of a set of paired binary masks and USI as shown in Fig 4.

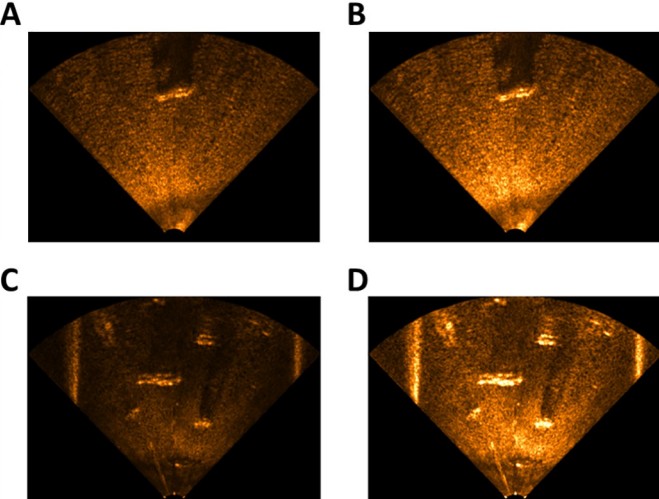

**Fig 7. Four datasets created in two environments.** (A) Reservoir low sensitivity, (B) Reservoir high sensitivity, (C) Pool low sensitivity, (D) Pool high sensitivity.

**Table 1. Hyperparameters of the image translation neural network.**

|  | *G* | *D* |
|---|---|---|
| Learning rate | 0.0004 | 0.0004 |
| Batch size | 1 | 1 |
| $\beta_1$ | 0.5 | 0.5 |
| $\beta_2$ | 0.999 | 0.999 |

Learning rate, batch size, $\beta_1$ and $\beta_2$ are the hyperparameters of the sub-NNs, *G* and *D*.

**Table 2. Hyperparameters of the semantic segmentation neural network.**

|  | *FCN* |
|---|---|
| Learning rate | 0.001 |
| Batch size | 5 |
| $\beta_1$ | 0.9 |
| $\beta_2$ | 0.999 |

Learning rate, batch size, $\beta_1$ and $\beta_2$ are the hyperparameters of FCN.

There were 200 paired images, each having a size of $640 \times 480$. The following table shows the description of the eight combined datasets used to train the semantic segmentation NN.

The semantic segmentation NN was trained using the eight training datasets with Adam optimizer for 200 epochs, and the hyperparameters were shown in Table 2.

Based on the eight datasets Table 3, we constructed three training datasets Table 4 to validate the effectiveness of the data augmentation using the synthetic USI, which were created using the image translation NN.

The T_Real dataset is a combination of the real datasets of the four conditions, and T_Synth is a combination of the synthetic datasets of the four conditions. On the other hand, T_Aug is a combination of the real datasets and synthetic datasets. When the semantic segmentation NN was trained using these datasets, the same number of paired images was used through uniform sampling. When the semantic segmentation NN was trained using the three

**Table 3. Real and synthetic underwater sonar image datasets.**

| Name | Environment | Sensitivity | Real/Synthetic |
|---|---|---|---|
| RHS_Real | Reservoir | High | Real |
| RHS_Synth | Reservoir | High | Synthetic |
| RLS_Real | Reservoir | Low | Real |
| RLS_Synth | Reservoir | Low | Synthetic |
| PHS_Real | Pool | High | Real |
| PHS_Synth | Pool | High | Synthetic |
| PLS_Real | Pool | Low | Real |
| PLS_Synth | Pool | Low | Synthetic |

RHS, reservoir high sensitivity; RLS, reservoir low sensitivity; PHS, pool high sensitivity; PLS, pool low sensitivity; _Real, real underwater sonar image; _Synth, synthetic underwater sonar image.

**Table 4. Real, synthetic and augmented underwater sonar image datasets.**

|  | Combinations |
|---|---|
| T_Real | RLS_Real + RHS_Real + PLS_Real + PHS_Real |
| T_Synth | RLS_Synth + RHS_Synth + PLS_Synth + PHS_Synth |
| T_Aug | T_Real + T_Synth |

T_Real, dataset is a combination of the real datasets of the four conditions; T_Synth, dataset is a combination of the synthetic datasets of the four conditions; T_Aug, dataset is a combination of the real datasets and synthetic datasets.

training datasets, the hyperparameters used were the same as those used when the eight training datasets shown in Fig 2 were used.

## Experimental results

### Synthetic underwater sonar image generation

Fig 8 shows samples of the results of generating the synthetic USI similar to the real USI of the four conditions using the model for the image translation NN.

Fig 8 confirms that the image translation NN can generate a synthetic USI by reflecting the characteristics of the real USI. The bottoms of the reservoir and pool are darker than the underwater objects. The underwater objects are brighter than the bottoms of the reservoir and pool, whereas there are brightness differences and shadows depending on the location of the sonar. Furthermore, when Fig 8A and 8C are compared with Fig 8B and 8D, it is observed that the effect of the sonar sensitivity on the USI can be determined from the image translation NN.

### Underwater sonar image segmentation

Fig 9 shows samples of the results of segmenting a real USI after training the semantic segmentation NN using the synthetic USI created by the underwater image translation (UIT) NN. Fig 9A–9C show that the pixels corresponding to the positions of the objects were segmented properly. However, in the case of Fig 9D, which is a result of segmenting a USI collected by increasing the sonar's sensitivity in the pool environment, it is observed that the pixels corresponding to the positions of the objects were not segmented properly, and the pixels corresponding to some parts of the pool boundary were segmented incorrectly.

We used two indicators called mean accuracy and mean intersection over union (IoU) to analyze the results of training the semantic segmentation NN quantitatively using the datasets created using the synthetic USI. Table 5 shows the results of segmenting the real USI with the semantic segmentation NN trained using the real USI datasets and synthetic USI datasets.

The performance values of the semantic segmentation NN trained using the real USI datasets and the synthetic USI datasets are shown in Table 5. This shows that there is no significant difference in performance between the semantic segmentation NNs trained using the synthetic USI datasets and those trained using the real USI datasets.

### Data augmentation results

We conducted an experiment to investigate whether the performance of the semantic segmentation NN can be improved when the synthetic USI generated by the proposed image translation NN is used for data augmentation. As shown in Fig 10, we used the dataset that contained both real USI and synthetic USI (T_Aug) and the dataset that contained only the real USI

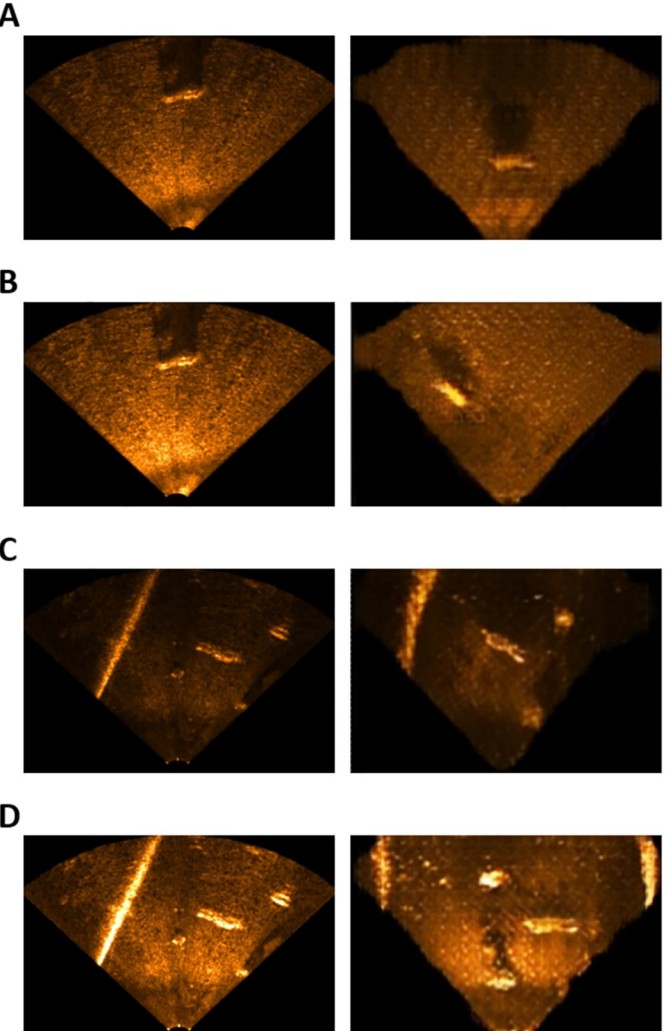

**Fig 8. Results of synthetic underwater sonar image generation (left: Real underwater sonar image, right: Synthetic underwater sonar image).** (A) Reservoir low sensitivity, (B) Reservoir high sensitivity, (C) Pool low sensitivity, (D) Pool high sensitivity.

(T_Real) to train the semantic segmentation NN, and then compared the results of segmenting real USIs. Table 6 shows the results of a quantitative analysis performed using the mean accuracy and mean IoU.

As shown in Table 6 the semantic segmentation NN trained using the real USI datasets and synthetic datasets and the semantic segmentation NN trained using the synthetic USI can segment the real USI properly.

The semantic segmentation NN trained using both the real USI and synthetic USI showed improved performance over that trained using only the real USI. However, when the real USIs collected in the pool environment (PLS_Real and PHS_Real) were segmented, the mean IoUs decreased slightly although the mean accuracies increased. Considering the characteristics of the mean IoU calculation process, we concluded that this occurred because some pixels of the USI were incorrectly segmented.

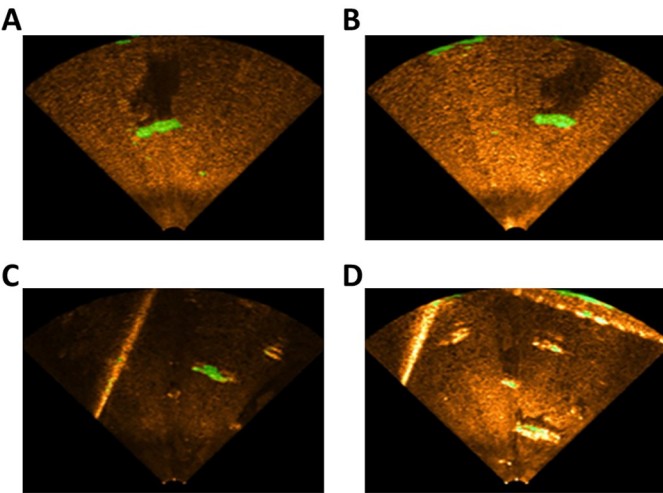

**Fig 9. Results of segmenting a real USI with the USI NN trained using the synthetic USI created by the PLS_Synth model.** (A) Reservoir low sensitivity, (B) Reservoir high sensitivity, (C) Pool low sensitivity, (D) Pool high sensitivity.

During the experiment, we used the same number of USI and binary mask pairs in T_Real, and T_Aug for the performance comparison. Nonetheless, when the generation of synthetic USI using the proposed UIT is applied in practice, the performance of the semantic segmentation NN will be improved because the proposed method can generate a large number of synthesized USIs for data augmentation.

## Conclusions and future works

In this paper, we proposed a data augmentation method using a UIT NN that can improve the semantic segmentation performance for object recognition in underwater environments where data collection is limited. The UIT is able to generate synthesized USIs with various poses and lighting conditions. If the proposed data augmentation method using the UIT is used, a large amount of synthetic USIs that are similar to real USIs can be created to train the semantic segmentation NN, even if the size of the real USI data is small. S1 Fig is the qualitative semantic segmentation result of Tables 5 and 6. S1 and S2 Tables are quantitative semantic

**Table 5. Semantic segmentation results using only one type of the real underwater sonar image datasets and synthetic underwater sonar image datasets.**

| Train dataset | Test dataset | Mean accuracy | Mean IoU |
|---|---|---|---|
| RLS_Real | RLS_Real | 0.77 | 0.75 |
| RLS_Synth | | 0.76 | 0.63 |
| RHS_Real | RHS_Real | 0.96 | 0.88 |
| RHS_Synth | | 0.81 | 0.71 |
| PLS_Real | PLS_Real | 0.87 | 0.76 |
| PLS_Synth | | 0.59 | 0.57 |
| PHS_Real | PHS_Real | 0.82 | 0.73 |
| PHS_Synth | | 0.81 | 0.71 |

Training dataset: real underwater sonar images and synthetic underwater sonar images.

Results of segmenting a real underwater sonar image with the semantic segmentation neural network trained using only one type of the real underwater sonar image datasets and synthetic underwater sonar image datasets.

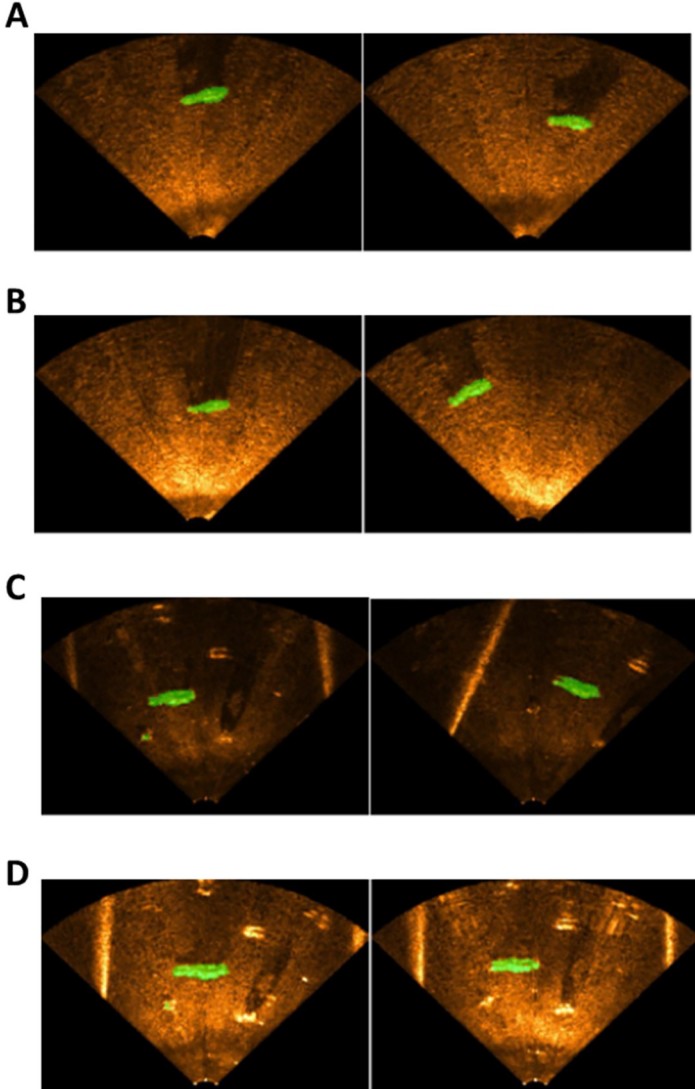

**Fig 10. USI segmentation results (left: Semantic segmentation NN trained using real USI, right: Semantic segmentation NN trained using real USI and synthetic USI).** (A) Reservoir low sensitivity, (B) Reservoir high sensitivity, (C) Pool low sensitivity, (D) Pool high sensitivity.

segmentation results that extend Tables 5 and 6. S1 Fig shows the results of training the semantic segmentation NN using the single types of training datasets separately. If the data used for training and testing have the same 'Environment' and 'Sensitivity' among the data types, the performance of Semantic Segmentation is good. On the other hand, when the types of data were different, the results of semantic segmentation was that the object of interest could not be recognized or the part that was not the object of interest was recognized as an object. These results are obtained when training with single type data, and when augmented data as shown in Table 4 is used for training, image segmentation performance is improved as shown in S1 Fig.

Additionally, if the semantic segmentation NN is trained using both the real USIs and synthetic USIs, the performance of semantic segmentation can be improved compared with when

**Table 6. Semantic segmentation results using a training dataset of real underwater sonar images and synthetic underwater sonar images.**

| Train dataset | Test dataset | Mean accuracy | Mean IoU |
|---|---|---|---|
| T_Real | RLS_Real | 0.75 | 0.73 |
| T_Aug | | 0.77 | 0.74 |
| T_Real | RHS_Real | 0.77 | 0.75 |
| T_Aug | | 0.84 | 0.76 |
| T_Real | PLS_Real | 0.74 | 0.71 |
| T_Aug | | 0.80 | 0.70 |
| T_Real | PHS_Real | 0.75 | 0.72 |
| T_Aug | | 0.83 | 0.69 |

Results of segmenting a real underwater sonar image with the semantic segmentation neural network trained using the real underwater sonar image datasets and synthetic underwater sonar image datasets.

it is trained using only the real USIs. Furthermore, as shown in S1 Fig, S1 and S2 Tables, semantic segment results are good even when using only synthetic USIs when training models, showing the possibility that only synthetic USIs can be used as training data.

The UIT NN used in this paper has the limitation that it needs explicit pairs of binary masks and USI. To mitigate this limitation, in the future, we can use a category-level UIT NN with cycle consistency loss is applied [24]. Furthermore, as the UIT NN used in this paper can generate a synthetic USI for one environment or condition, the parameters of the NN have to be trained again to generate a synthetic USI for another environment or condition. To mitigate this, we can use a UIT NN that can handle multiple domains at once [25].

## Supporting information

**S1 Fig. Results of image segmentation by type of datasets (horizontal axis: Type of test datasets, vertical axis: Type of training datasets).** (A) Reservoir low sensitivity_Real, (B) Reservoir high sensitivity_Real, (C) Pool low sensitivity_Real, (D) Pool high sensitivity_Real, (E) Reservoir low sensitivity_Synth, (F) Reservoir high sensitivity_Synth, (G) Pool low sensitivity_Synth, (H) Pool high sensitivity_Synth, (I) T_Real, (J) T_Synth, (K) T_Aug.
(TIF)

**S1 Table. Mean accuracy results of image segmentation by type of datasets (horizontal axis: Type of test datasets, vertical axis: Type of training datasets).** (A) Reservoir low sensitivity_Real, (B) Reservoir high sensitivity_Real, (C) Pool low sensitivity_Real, (D) Pool high sensitivity_Real, (E) Reservoir low sensitivity_Synth, (F) Reservoir high sensitivity_Synth, (G) Pool low sensitivity_Synth, (H) Pool high sensitivity_Synth, (I) T_Real, (J) T_Synth, (K) T_Aug.
(TXT)

**S2 Table. Mean IoU results of image segmentation by type of datasets (horizontal axis: Type of test datasets, vertical axis: Type of training datasets).** (A) Reservoir low sensitivity_Real, (B) Reservoir high sensitivity_Real, (C) Pool low sensitivity_Real, (D) Pool high sensitivity_Real, (E) Reservoir low sensitivity_Synth, (F) Reservoir high sensitivity_Synth, (G) Pool low sensitivity_Synth, (H) Pool high sensitivity_Synth, (I) T_Real, (J) T_Synth, (K) T_Aug.
(TXT)

**S3 Table. Abbreviation table.**

(TXT)

**S1 Dataset.**

(Z01)

**S2 Dataset.**

(ZIP)

## Author Contributions

**Conceptualization:** Byungjae Park, Ayoung Kim, Sejin Lee.

**Data curation:** Eon-ho Lee, Byungjae Park.

**Formal analysis:** Byungjae Park, Ayoung Kim, Sejin Lee.

**Investigation:** Eon-ho Lee, Myung-Hwan Jeon, Hyesu Jang.

**Writing – original draft:** Eon-ho Lee, Byungjae Park, Sejin Lee.

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
