## [Decision Letter · Decision Letter 0]

26 May 2022

PONE-D-22-11679Data Augmentation Using Image Translation for Underwater Sonar Image SegmentationPLOS ONE

Dear Dr. LEE,

Thank you for submitting your manuscript to PLOS ONE. After careful consideration, we feel that it has merit but does not fully meet PLOS ONE’s publication criteria as it currently stands. Therefore, we invite you to submit a revised version of the manuscript that addresses the points raised during the review process.

We look forward to receiving your revised manuscript.

Kind regards,

Mahdi Abbasi, PhD.

Academic Editor

PLOS ONE

Journal Requirements:

"This research was supported by the National Research Foundation of Korea (NRF)

grant funded by the Korean government (MSIT) (No. 2019R1F1A1053708,

2021R1F1A1057949) and ”Regional Innovation Strategy (RIS)” through the National

Research Foundation of Korea(NRF) funded by the Ministry of Education(MOE)

(2021RIS-004)."

"This research was supported by the National Research Foundation of Korea (NRF) grant funded by the Korean government (MSIT) (No. 2019R1F1A1053708, 2021R1F1A1057949) and "Regional Innovation Strategy (RIS)" through the National Research Foundation of Korea(NRF) funded by the Ministry of Education(MOE) (2021RIS-004)."

Reviewers' comments:

Reviewer's Responses to Questions

**Comments to the Author**

1. Is the manuscript technically sound, and do the data support the conclusions?

Reviewer #1: Partly

Reviewer #2: Partly

2. Has the statistical analysis been performed appropriately and rigorously? 

Reviewer #1: Yes

Reviewer #2: Yes

3. Have the authors made all data underlying the findings in their manuscript fully available?

Reviewer #1: No

Reviewer #2: Yes

4. Is the manuscript presented in an intelligible fashion and written in standard English?

Reviewer #1: Yes

Reviewer #2: Yes

5. Review Comments to the Author

Reviewer #1: In this manuscript, the authors used the Pix2Pix Generative Adversarial Network for image translation and then leverage it for data augmentation. Then a Fully Convolutional Network is used for image segmentation. The main aim of this research is to avoid over-fitting the model. Due to the extensive research conducted recently, it is absolutely necessary to pay attention to the following.

1- I suggest you have a brief overview of the related work on data augmentation and image segmentation. You can transfer a subsection of the "Introduction" to the "Related work" section.

2- The whole manuscript is full of abbreviations without defining them. Please define them before use.

3- Please, describe the details of the neural network architecture used. Includes the number of layers, neurons and activation functions and so on.

4- Please, describe in detail the data used. Includes the number of real data, synthetic data, dimensions of images and so on.

5- Please describe how to combine synthetic and real data to make augmented data.

6- What is the advantage of using FCN? That convinces us to augment data to learn the model.

7- Please provide a comparison without the use of incremental data to show that your method prevents over-fitting

Reviewer #2: This paper has been to improve Underwater Sonar Image Segmentation by data augmentation and reduce the limitations of previous methods. However, before publishing the paper, I suggest some minor revisions, as follows:

1. In the abstract, please mention the exact amount of improvement and the metric that has been improved.

2. Organize the article in a standard format. Some sections, including sections 3 and 4, can be merged into one section.

3. It is necessary to provide an architecture of the proposed method in your work.

4. In the result section, various tables have been provided. However, it is not discussed comprehensively.

6. PLOS authors have the option to publish the peer review history of their article (what does this mean?). If published, this will include your full peer review and any attached files.

Reviewer #1: **Yes: **Amin Nazari

Reviewer #2: **Yes: **Fazeleh Tavassolian

---

## [Author Response · Author response to Decision Letter 0]

8 Jul 2022

We would like to thank you and the reviewers for thorough reading of our manuscript and the comments which helped us to enhance the quality of the work. Their constructive comments are well-received and highly appreciated. We have carefully considered the reviewers' recommendation to further improve the clarity and quality of our manuscript. We hope that the revision made have improved the manuscript at all levels, and that the changes made in an attempt to address the comments of the reviewers are satisfactory. Detailed information is attached as a rebuttal letter file.

---

## [Decision Letter · Decision Letter 1]

25 Jul 2022

Data Augmentation Using Image Translation for Underwater Sonar Image Segmentation

PONE-D-22-11679R1

Dear Dr. LEE,

We’re pleased to inform you that your manuscript has been judged scientifically suitable for publication and will be formally accepted for publication once it meets all outstanding technical requirements.

Kind regards,

Mahdi Abbasi, PhD.

Academic Editor

PLOS ONE

Additional Editor Comments (optional):

Reviewers' comments:

Reviewer's Responses to Questions

**Comments to the Author**

1. If the authors have adequately addressed your comments raised in a previous round of review and you feel that this manuscript is now acceptable for publication, you may indicate that here to bypass the “Comments to the Author” section, enter your conflict of interest statement in the “Confidential to Editor” section, and submit your "Accept" recommendation.

Reviewer #1: All comments have been addressed

Reviewer #2: All comments have been addressed

2. Is the manuscript technically sound, and do the data support the conclusions?

Reviewer #1: Yes

Reviewer #2: Yes

3. Has the statistical analysis been performed appropriately and rigorously? 

Reviewer #1: Yes

Reviewer #2: Yes

4. Have the authors made all data underlying the findings in their manuscript fully available?

Reviewer #1: No

Reviewer #2: Yes

5. Is the manuscript presented in an intelligible fashion and written in standard English?

Reviewer #1: Yes

Reviewer #2: Yes

6. Review Comments to the Author

Reviewer #1: (No Response)

Reviewer #2: The authors have clarified several of the questions I raised in my previous review. The title and abstract are appropriate for the content of the text. Furthermore, the article is well constructed, the experiments were well conducted, and analysis was well performed. The conclusion presented by this manuscript seems correct.

7. PLOS authors have the option to publish the peer review history of their article (what does this mean?). If published, this will include your full peer review and any attached files.

Reviewer #1: No

Reviewer #2: **Yes: **Fazeleh Tavassolian

---

## [Editor Report · Acceptance letter]

4 Aug 2022

PONE-D-22-11679R1 

Data Augmentation Using Image Translation for Underwater Sonar Image Segmentation 

Dear Dr. LEE:

I'm pleased to inform you that your manuscript has been deemed suitable for publication in PLOS ONE. Congratulations! Your manuscript is now with our production department. 

Kind regards, 

on behalf of

Dr. Mahdi Abbasi 

Academic Editor

PLOS ONE